# Insomnia among Cancer Patients in the Real World: Optimising Treatments and Tailored Therapies

**DOI:** 10.3390/ijerph20053785

**Published:** 2023-02-21

**Authors:** Irene Pinucci, Annalisa Maraone, Lorenzo Tarsitani, Massimo Pasquini

**Affiliations:** Department of Human Neurosciences, Sapienza University of Rome, 00185 Rome, Italy

**Keywords:** insomnia, insomnia symptoms, cancer, cancer related insomnia, pharmacologic treatment

## Abstract

Background: Insomnia is commonly reported by cancer patients. Its multifaceted pathophysiology makes this symptom a complex challenge for the clinician, who should bear in mind the manifold world of causes and consequences of sleep disturbances in these patients and the importance of accurate treatment that should consider the frequent co-prescription of multiple medications. With our work, we aim to provide a tool to better master the treatment of this symptom in cancer patients, considering the gap between clinical and pharmacodynamic knowledge about the efficacy of different molecules and evidence-based prescribing. Methods: A narrative review of the studies investigating the pharmacological treatment of insomnia in cancer patients was conducted. Three hundred and seventy-six randomised controlled trials (RCTs), systematic reviews and meta-analyses were identified through PubMed. Only publications that investigated the efficacy of the pharmacological treatment of insomnia symptoms in cancer patient were considered. Results: Among the 376 publications that were individuated, fifteen studies were eligible for inclusion in the review and were described. Pharmacological treatments were outlined, with a broad look at specific clinical situations. Conclusions: The management of insomnia in cancer patients should be personalised, as is already the case for the treatment of pain, taking into account both the pathophysiology and the other medical treatments prescribed to these patients.

## 1. Introduction

### 1.1. Epidemiology of Insomnia in Cancer Patients

Insomnia is unfortunately one of many factors that worsen the quality of life of cancer patients, and numerous studies have documented its high frequency. Insomnia symptoms have been described in nearly half the patients who have received a recent cancer diagnosis. Severe sleep difficulties have been reported by a wide range from 25 to 59% of cancer patients, double the rate described in the general population [1]. A recent systematic review of sleep disturbances and/or sleep disorders in cancer found a prevalence up to 95% in these patients [2]. Differences in the observed rates may depend on heterogeneity in the definition and measurement of insomnia symptoms [3]. Studies using objective measurements of sleep quality in cancer patients are scarce, but actigraphic measurements of sleep outcomes in cancer patients undergoing chemotherapy show an aggravation of sleep disturbances with continuation of the chemotherapy regimen through several series [4]. A recent study reported the presence of clinically significant sleep difficulties in 64% of patients with cancer, even though only a small portion of them mentioned such disorders as their first concerns in their integrative oncology consultation [5]. Despite the frequency of insomnia in cancer patients, its consequences on daytime functioning and health are often overlooked. Studies focus on the various stages of the disease by evaluating the consequences of diagnosis and therapy on sleep, also assessing the consequences on sleep quality after a survival period of months or years. Some authors focused on objective measurements (through actigraphy and polysomnography) of sleep quality in breast cancer patients before the start of chemotherapy, describing disturbed sleep and fatigue prior to the treatment and suggesting a role for early intervention [6]. Insomnia symptoms were also investigated during the chemotherapy treatment, noting that, on the seventh day following the first chemotherapy cycle, 36.6% of patients reported insomnia symptoms, and 43% met the diagnostic criteria for insomnia syndrome, a number three times higher than in the general population [7]. An 18-month longitudinal study on 856 patients with heterogeneous cancer sites and stages showed a general decline of the prevalence of insomnia, even though its rates were still considerable at the end of the study [8]. Finally, daytime sleepiness and sleep duration were investigated in long-term cancer survivors. While no association was found between a history of cancer and sleep duration, daytime sleepiness was found to persist in individuals diagnosed more than two years earlier [9]. Another study focused on 1–10-year breast cancer survivors describing severe subjective insomnia [10]. Concerning the cancer site, breast, gynaecologic, and lung cancers appear to result in the highest risk of developing insomnia symptoms [7,10,11], while lower rates were found in men with prostate cancer [8]. Among all cancer sites, breast cancer has the highest prevalence of insomnia symptoms [6,7,11,12], and most studies on disturbed sleep in cancer patients focus on women with breast cancer [13,14,15]. This prevalence could be linked to several causes: nocturnal awakenings are related to the hot flashes caused by treatment [16,17], and, as also mentioned for other cancer sites, pain and stress [18,19] play a major role.

### 1.2. Pathophysiology of Insomnia in Cancer Patients

The causes and effects of insomnia in cancer patients may have a mutual causal relationship. Psychiatric diseases, particularly anxiety and depressive disorders, are frequent comorbidities in cancer patients [13,20,21]. Insomnia is a frequently present symptom for both these disorders. Demoralisation, often presented by cancer patients, may accompany mood disorders and has been linked to increased sleep difficulties in breast cancer patients [22,23,24]. However, it is unlikely that the high prevalence of insomnia symptoms in cancer patients is exclusively related to psychiatric symptoms [1]. During this challenging pandemic period, the prevalence rates of psychiatric symptoms and insomnia are significantly raised [25], and we may expect negative consequences, especially in cancer patients, who may find it more difficult to access care services and cancer treatments. Stress-related cancer diagnosis (Distress) is associated with hyperarousal, a state of increased somatic, cortical, and cognitive activation, and some data have shown increased cortisol levels, body temperature, 24-h metabolic rate, and heart rate in cancer patients with insomnia symptoms. In addition, pain provoked by the disease or its surgical or pharmacological treatment may easily impact sleep quality [2,26]. Pain is, in fact, one of the most disabling symptoms for the cancer patient, affecting more than 50% of patients at any stage of the disease and more than two-thirds of those with metastatic or advanced disease [27]. Numerous studies have revealed that people with severe or chronic pain have a higher prevalence of depressive symptoms and insomnia [2]. In addition, the close relationship between pain and depression may also play a reciprocal role in amplifying the effects on insomnia. The pain–depression binomial would seem to have a dual reinforcing mechanism as its core. On the one hand, untreated pain is linked to a decline in psychological defence mechanisms, predisposing patients toward the emergence of psychological disorders. On the other hand, depressive symptoms may increase sensitivity and perception of pain, lowering the threshold and amplifying sufferance [28].

Moreover, as many treatments prescribed for noncancer patients can result in sleep disturbances [29], this is also true for anticancer drugs. Treatments can be a cause of sleep disturbances either for the provoked emotional distress or for their direct side effects. Breast cancer patients and survivors are prone to have insomnia caused by the aforementioned hot flashes but also by the consequences of chemotherapy, radiotherapy therapy, and hormone therapy [1,17,30,31]. Indeed, radiotherapy and chemotherapy (this one in particular) have been associated with sleep disturbances. Multiple side effects of these treatments may worsen insomnia symptoms. The main cause could be reported by the impact of these agents on body functions and effects such as pain, diarrhoea, nausea, and vomiting [31]. Glucocorticoids, often prescribed in the supportive care of cancer patients, could result in an alteration of the sleep–wake cycle through disruption of the cortisol rhythm [32].

One study focused on the molecular mechanisms associated with sleep disruption in cancer patients, suggesting a reciprocal causative approach, a “‘chicken or the egg’ phenomenon”, where poor sleep takes a role in tumorigenesis and cancer progression [33]. These findings are consistent with recent studies suggesting that chronic circadian disruption (caused, for example, by a job requiring night shifts) has a causative effect on the pathophysiology of breast cancer and its metastatic dissemination [34]. Moreover, sleep efficiency as measured by actigraphy is related to prognostic outcomes in patients with advanced breast cancer [35]. Studies investigating how insomnia symptoms may be both a risk factor and a consequence of cancer, focusing on many different cancer sites, have been described in a recent comprehensive review [36]. Many authors refer to a behavioural model for sleep disturbances called the “3-P’s Model”, first described in 1987 [37,38], which describes the different causes as consisting of three groups: predisposing, precipitating, and perpetuating factors. While predisposing factors are represented by sex (women tend to have more sleep disturbances than men), anxiety traits, or the presence of a psychiatric disorder and a family or personal history of insomnia, precipitating factors are more related to the consequences of cancer and its treatments, such as pain and side effects of drugs and surgery. Finally, perpetuating factors involve maladaptive sleep behaviours as a shifting sleep phase or spending more time in bed or, for example, not following sleep hygiene recommendations [9,30,39,40,41,42,43]. Investigating and treating insomnia symptoms in cancer patients is critical, especially considering their comorbidity with fatigue, a frequent consequence of cancer and cancer treatments. When approaching a patient reporting cancer-related fatigue or excessive daytime sleepiness (EDS) [44], its relationship with cancer-related sleep disorders should be appropriately evaluated [11,45]. Along with and before the pharmacological approach to insomnia symptoms, nonpharmacological treatment should be considered by the clinician. First, sleep hygiene is an important tool for the treatment of insomnia in cancer patients, who tend to spend more time in bed during the day, with consequences on the sleep–wake schedule caused by frequent naps [1]. Therefore, a sleep diary and sleep hygiene education can be useful tools for the initial approach to insomnia symptoms in cancer patients, as in the general population. The American Sleep Association has listed some basic tips for improving sleep quality, including going to bed at a regular time; avoiding naps during the day; leaving the bed if not asleep after 5–10 min; avoiding watching TV or reading in bed; avoiding caffeinated beverages in the afternoon; avoiding smoking, alcohol, and over-the-counter sleep medication; exercising daily but not before bedtime; and, finally, having a quiet, comfortable room and a prebedtime routine [46]. A combination of pharmacological and nonpharmacological treatments could be a useful strategy to “break the vicious cycle of fatigue and insomnia” often presented by cancer patients [26]. Nonpharmacologic treatments such as CBT have been shown to be an effective approach, but their description is beyond the scope of this study. If it is ultimately decided to prescribe drug therapy for the treatment of insomnia in a cancer patient, many considerations are necessary. These include hepatic metabolism and the resulting pharmacokinetic interactions. A classic example of a pharmacokinetic interaction between psychiatric drugs and chemotherapeutics is the effect that SSRIs have on cytochrome CYP2D6 and the consequences for tamoxifen metabolism, reducing its clinical benefits [47]. These interactions may concern the relationship between other psychotropic drugs and chemotherapeutic agents [48,49]. Some of these interactions will be described below.

### 1.3. Objectives

Although several studies have already addressed the effectiveness of pharmacological and nonpharmacological treatments of insomnia symptoms, a lack of evidence has emerged in the context of cancer patients. There is a lack of clinical guidance that considers the efficacy and safety of pharmacological treatments in these patients. With this in mind, we sought to narratively review the literature to describe the various options available for the pharmacological treatment of insomnia symptoms in cancer patients, focusing on the main indications, efficacy, and response to the treatments and evaluating potential and reliable alternatives based on specific insomnia symptoms.

## 2. Materials and Methods

We conducted a literature review of the most significant studies regarding the pharmacological treatment of insomnia symptoms in patients with cancer. We searched the PubMed database for randomised clinical trials (RCTs), systematic reviews, and meta-analyses published starting from January 1990 to January 2022 on the abovementioned topic. The literature search was conducted in December 2022. We employed the following keywords: insomnia AND cancer AND (“pharmacological treatment” OR “drug treatment” OR “psychotropic” OR “medication”). The 376 title publications were screened, and only publications that investigated the drug treatment of insomnia in cancer patients were considered for inclusion. All works written in English that considered other sleep disorders instead of insomnia were excluded. Works that evaluated the effectiveness of nonpharmacological treatments, e.g., CBT or others (yoga, mindfulness) were not included. Articles assessing the rise of insomnia in the pathogenesis of cancer were also excluded. Forty-seven articles were read, and fifteen publications were included in the present study.

## 3. Results

Table 1 shows the fifteen publications included in the study.

## 4. Discussion

Despite decades of discussion around what characteristics an ideal sleeping pill should have, we are still far from finding a medication that is safe, effective, does not cause tolerance or dependence, and induces a sleep pattern comparable to natural sleep [62]. We believe that the complexity of the cancer patient necessitates a range of clinical and pharmacological knowledge in order to find the most appropriate resource for cancer patients. Therefore, the goal of our work was to identify which medications have already been described for the treatment of insomnia in cancer patients, regardless of whether they are provided for in the guidelines for insomnia or have this symptom among their indications. The fifteen publications that were included in this review took into account cancers from various sites and medications of different pharmacological categories. We will therefore go into further detail about the useful information to be borne in mind of the drugs that feature most prominently in the included publications.

Benzodiazepine and non-benzodiazepine hypnotics are the most frequently used medications for the treatment of insomnia in cancer patients as well. Recent Italian and international guidelines for the treatment of insomnia tend to suggest the prescription of short-term (less than four weeks) use GABAa receptor agonist compounds (short-to-medium half-life benzodiazepine and non-benzodiazepine compounds) for the treatment of acute insomnia and as a second line after CBT-I for chronic insomnia [43,62,63,64,65]. However, these compounds have been firmly contraindicated in patients older than 65 years old [66]. In addition, there is a lack of guidelines specifically dedicated to cancer patients. Although studies upon their efficacy among these patients are scarce, seven of the studies included in this review described the use of these drugs in cancer patients with carcinomas of different origins [1,14,15,51,52,56,59]. In addition to the effects on insomnia, these compounds demonstrated antiemetic properties, and their amnesic properties may help patients coping with conditioning mechanisms related to anticipatory nausea and vomiting (ANV) [32]. In cases where cancer involves the central nervous system (CNS), benzodiazepines can help control seizures. Benzodiazepines that are only metabolised by glucuronidation not having active metabolites are preferable when there is a hepatic involvement (for instance, lorazepam or oxazepam) [32]. These compounds can cause the well-known physical dependence and physical tolerance that can be prevented with a short time prescription (less than 4 weeks) and gradual dosage reduction. However, in advanced stages of the disease, concern about dependence should not prevent their prescription. Benzodiazepine can also alter the sleep architecture, increasing the N2 stage and decreasing the N3 and REM stages of sleep [67]. The Zolpidem augmentation of treatment with Venlafaxine and SSRI has shown to improve sleep and quality of life in breast cancer survivors [68].

Melatonin (especially in its 2 mg extended-release format) is another first-line treatment for insomnia that has also been found to be safe even in older populations [69,70]. Seven studies that discussed the usage of these medications in cancer patients were selected [54,56,59,60,61]. In Italy, as in other countries, its indications include the monotherapy of insomnia in people older than 55 years. Its efficacy has also been studied in patients undergoing the first cycle of adjuvant chemotherapy and in perioperative breast cancer patients [71,72]. This compound would have numerous benefits in addition to promoting and maintaining sleep. It would, in fact, have a beneficial effect on the toxicity of some chemotherapeutic drugs [73,74]. In addition, many studies have investigated the anticancer properties of this hormone [75,76,77]. Melatonin is metabolised by CYP1A: Fluvoxamine, hormone replacement therapy (oestrogens), and cimetidine could increase its blood levels.

Considering the frequent comorbidity of mood disorders in cancer patients, antidepressants are useful compounds both for their efficacy on the frequent comorbid depressive symptoms and, in some cases, for their sedative effects [78,79]. Of the articles included in this paper, three described the use of tricyclic drugs [51,56,59], four described the use of trazodone [15,51,56,59], one described the use of venlafaxine [55], and, finally, four considered mirtazapine for its hypnoinductive capabilities in cancer patients [15,26,51,59].

Amitriptyline (tricyclic antidepressant with H1, 5HT2, and cholinergic muscarinic antagonism) and trazodone (serotonin antagonist and reuptake inhibitor (SARI) with 5HT1a, 5HT2, and alpha1 antagonist actions) are frequently used for the treatment of insomnia in Italy despite the scarce evidence of their efficacy [43,80] both in the general population and in cancer patients [15,32,51,56,59]. International guidelines still suggest antidepressant prescriptions for the treatment of insomnia in the general population in the presence of a coexistent mood disorder. Anticholinergic side effects of amitriptyline, especially constipation, should be kept in mind in patients taking opioids. Moreover, as other TCAs, amitriptyline could inhibit CYP2C19 threatening the efficacy of cyclophosphamide [32], and its administration with Oxaliplatin could lead to an increased risk of QT interval prolongation [49]. Moreover, hormone replacement therapy (oestrogens), cimetidine (inhibiting CYP1A), fluvoxamine, fluoxetine, paroxetine, bupropion, and duloxetine (inhibiting CYP2D6) could increase its blood levels.

Mirtazapine, a noradrenergic and specific serotonergic antidepressant (NaSSA) with action on the alpha2, 5HT2, 5HT3, and H1 receptors, is able to result in sleep induction and a longer sleep duration compared to benzodiazepines. Its efficacy in treating nausea, weight loss (through increased appetite), and hot flashes makes mirtazapine an effective treatment for multiple distressing symptoms of cancer [15,32,50,51,59,81,82,83,84]. Mirtazapine is characterised by scarce pharmacokinetics interactions.

Some antipsychotics, given their sedative and antiemetic actions, are effective compounds for the treatment of cancer-related symptoms alone or in addition to antidepressants for treatment-resistant depression. Three of the articles included in this paper discussed the use of olanzapine [53,57,58], while one described the use of quetiapine for its capacity to improve insomnia symptoms in cancer patients [15]. Atypical antipsychotics, such as quetiapine and olanzapine, have a sedative effect and a well-known appetite-enhancing effect that could represent a desirable consequence in patients facing cancer-related weight loss [51,53,58]. Their efficacy in the treatment of chronic nausea and vomiting in cancer patients has been widely investigated [85,86,87]. Despite the scarce evidence of their efficacy for insomnia treatment, quetiapine and olanzapine are widely used as off-label treatments for sleeping difficulties in the general population and patients with comorbid medical conditions, such as cancer patients [15,80]. The efficacy of quetiapine has also been described for tamoxifen-induced insomnia in a sample of women with breast cancer [78]. No work has emerged that has considered this drug for the treatment of insomnia in cancer patients, haloperidol is a commonly prescribed compound for the treatment of nausea and vomiting in cancer patients [88,89], and, considering its sedative effects, its evening administration could increase its efficacy for both nausea and sleep difficulties. In addition to the metabolic effects of these compounds, other side effects such as extrapyramidal symptoms, orthostatic hypotension, QTc interval prolongation, arrhythmias, and haematological consequences should be considered. In particular, quetiapine with oxaliplatin could determine an increased risk of QT interval prolongation [49]. Furthermore, and particularly important in breast cancer patients, antipsychotics could also lead to hyperprolactinemia. Indeed, antipsychotics that increase prolactin may have a significant impact on the likelihood of developing breast and prostate cancer over time [90,91]. It is crucial to monitor prolactinemia in patients, although it should be borne in mind that the dosages used for hypnoinductive purposes alone may give this undesirable effect more rarely, and compounds such as quetiapine almost never seem to produce an increase of prolactin in the blood [92,93]. Finally, when patients have CNS cancer or take high doses of corticosteroids, it is essential to consider their epileptic threshold reduction [87]. Their association with prochlorperazine or metoclopramide could lead to an increased risk of akathisia and tardive dyskinesia, while methadone or ondansetron could increase the risk of QTc prolongation [30]. The above characteristics of the different compounds and their levels of evidence regarding prescription for insomnia in the general population and cancer patients are summarised in Table 2.

## 5. Conclusions

Insomnia among cancer patients is a very frequent issue, yet it is often misdiagnosed and undertreated, especially when it comes to outpatients. Sleep hygiene and CBT are both effective nonpharmacological treatments, especially when insomnia is not attributable to iatrogenic causes. Benzodiazepines are prescribed too often, considering their negative effects on sleep architecture and their consequences related to tolerance and dependence. Specialists often resort to the antihistaminergic effects of antidepressants and atypical antipsychotics at low dosages. However, these treatments need expertise and attention, especially considering the off-label nature of prescribing some of them, for example, compounds belonging to the antipsychotics class. In our opinion, the management of sleep disorders in cancer patients should be individualised, as is already done for the treatment of pain, considering both the pathophysiology and other medical treatments prescribed to these patients. The gap between clinical and pharmacodynamic knowledge about the efficacity of different molecules and the evidence-based prescriptions represent an important and often unknown feature of clinicians’ work. For this reason, we intended to provide a tool that took into account the characteristics of different compounds and their relationships to cancer comorbidities. Table 2 summarised the main characteristics of the most frequently prescribed compounds for the treatment of insomnia and highlighted their levels of evidence for both the general population and cancer patients. Such a tailored and broader view of the clinical presentation and treatment of insomnia could lead to breaking the vicious cycle that characterises the nature of a major problem that cancer patients often face.

## Figures and Tables

**Table 1 ijerph-20-03785-t001:** RCTs, systematic reviews, and meta-analyses on the pharmacological treatment of insomnia in cancer patients.

Author	Description	Pharmacological Treatment Described
Savard J, et al. [1]	Review	Hypnotics
Theobald DE, et al. [50]	RCT	Mirtazapine
O’Donnell JF. et al. [51]	Review	Hypnotics, TCAs, trazodone, mirtazapine, antihistamines, atypical antipsychotic agents
Fiorentino L, et al. [14]	Review	Hypnotics
Berger AM, et al. [52]	Review	Hypnotics
Tan L, et al. [53]	RCT	Olanzapine
Chen WY, et al. [54]	RCT	Melatonin
Ensrud KE, et al. [55]	RCT	Venlafaxine, Estradiol
Matthews EE, et al. [56]	Review	Hypnotics, TCAa, Ramelton, Trazodone, Melatonin
Davis MP, et al. [57]	Review	Olanzapine
Kwak A, et al. [15]	Review	Hypnotics, Trazodone, Mirtazapine, Quetiapine, Gabapentin, Ramelton, Antihistamines
Zhou JG, et al. [58]	Meta-Analysis	Olanzapine
Acker KA, et al. [59]	Review	Hypnotics, Ramelton, Suvorexant, Mirtazapine, Trazodone, TCAs, Gabapentin, Tiagabine, Melatonin, Diphenhydramine
Jafari-Koulaee A, et al. [60]	Review	Melatonin
Palagini L, et al. [61]	Review	Melatonin

**Table 2 ijerph-20-03785-t002:** Main characteristics of the most frequently prescribed compounds for the treatment of insomnia in cancer patients.

Medications	Classification	Most Common Side Effects	Pharmacokinetics	Clinical Useful Information
Lorazepam, Diazepam, Delorazepam.	Benzodiazepines	Sedation, memory issues, dizziness. Hypotension, decrease respiratory drive or respiratory arrest during intoxication.	Lorazepam is only metabolised via glucuronidation and has not active metabolites.	Short-term prescription (<4 weeks) to prevent physical dependence and tolerance. To avoid in patients over 65 yo. Choose lorazepam when hepatic injury.
Zolpidem, Zaleplon	Z-compounds	Sedation, memory issues, dizziness.	Metabolised by CYP3A4, 1A2, 2C9.	Short-term prescription (<4 weeks). To avoid in patients over 65 yo.
Melatonin		Headache, dizziness, nausea, drowsiness.	Metabolised by CYP1A: Fluvoxamine, hormone replacement therapy (oestrogens) and cimetidine could increase its blood levels.	Some evidence of its efficacy in preventing delirium in elderly patients.
Amitriptyline	Tricyclic	Sedation, orthostatic hypotension, anticholinergic effects. QTc prolongation.	Metabolised by CYP2D6, CYP1A2 (active metabolite nortriptyline via demethylation), 2C9, 2C19, 3A4. Fluvoxamine, hormone replacement therapy (oestrogens), cimetidine (inhibiting CYP1A) and fluoxetine, paroxetine, bupropion and duloxetine (inhibiting CYP2D6) could increase its blood levels. With Oxaliplatin, augmented risk of QTc prolongation	To avoid in patients taking opioids for its constipating effects.
Trazodone	SARI Antidepressant	Nausea, constipation, orthostatic hypotension, dizziness, sedation, SIADH, serotonin syndrome.	Metabolised by CYP3A4.	
Mirtazapine	NaSSA Antidepressant	Sedation, weight gain, dry mouth, dizziness.	Metabolised by CYP2D6 and 3A4. Scarce pharmacokinetics interactions.	Desirable weight gain in cancer patients, anti-nausea effect. Used for hot flashes.
Haloperidol	Typical antipsychotics	EPS, Hyperprolactinemia.	Metabolised by CYP1A2, 2D6, 3A4. With Prochlorperazine or Metoclopramide augmented risk of akathisia and tardive dyskinesia. With Methadone, Cisplatin or Ondansetron augmented risk of QTc prolongation.	Anti-nausea effect.
Olanzapine	Atypical antipsychotics	Weight gain, sedation, dizziness, risk augmentation for diabetes and dyslipidaemia, constipation.	Metabolised by CYP1A2, 2D6, 3A4. With Prochlorperazine or Metoclopramide augmented risk of akathisia and tardive dyskinesia. With Methadone or Ondansetron augmented risk of QTc prolongation.	Desirable weight gain in cancer patients.
Quetiapine	Atypical antipsychotics	Sedation, weight gain.	Metabolised by CYP3A4, 2D6. With Prochlorperazine or Metoclopramide augmented risk of akathisia and tardive dyskinesia. With Methadone, Oxaliplatin or Ondansetron augmented risk of QTc prolongation.	Desirable weight gain in cancer patients.

## Data Availability

The data that support the findings of this study are available from the corresponding author upon reasonable request.

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
