# Peer review of "Insomnia among Cancer Patients in the Real World: Optimising Treatments and Tailored Therapies"

_ijerph, 2023, doi:10.3390/ijerph20053785_

Round 1

Reviewer 1 Report

The article make reference to the management of insomnia in cancer patients which should be personalized.

The title should be reformulated. In my opinion, I find the expression ``in the real world`` not very scientific.  

Chapter Results – regarding table 1 there are 2 titles, please remove one of them.

Chapter Discussion: I suggest you to reorganized the Discussion by starting with the 15 articles included in this review, depending also on the objectives and materials and methods (chapter 1.3 and 2).

It was mentioned 15 publications in this study, but in discussion there is not any particular comments regarding these articles. Please explain why you choose these 15 articles and motived the clinical or pharmaceutical significance of all these articles.

Reviewer 2 Report

In their narrative review paper, the Authors evaluated the treatment option for patients with cancer related insomnia, analyzing fifteen out of 376 publications.

Even if the paper is well explained and clearly written, I have some major concerns that should be addressed.

Briefly, I suggest:

-          many and recent reviews about this topic can be found (also in IJERPH) so the specificity of the paper should be increased

-          in methods it is not clearly explained the criteria of inclusion/exclusion of paper selection (“The 376 title publications were 145 screened, and only publications that investigated the drug treatment of insomnia in cancer patients were considered for inclusion. Forty-seven articles were read, and 15 publications were included in the present study.”)

-          it is not debated the hepatic catabolism of melatonin and consequent interactions with other drugs (mainly chemotherapeutic agents)

-          as regards pharmacokinetics of drugs usually administered in patients with cancer, the reciprocal influences on CYP450 isoenzymes should be deeply analyzed, considering not only inhibition but also activation, as well as the substrate competition and/or specificity of these drugs

-          the effects of some drugs (e.g. antidepressants and antipsychotics) on prolactin incretion and their administration in patients with breast cancer should be discussed

-          the pharmacodynamics properties (and consequent side effects) and interactions of  drugs frequently used in these patients should be discussed

Round 2

Reviewer 1 Report

The work is semnificately improved

Reviewer 2 Report

The Authors well addressed the questions/suggestions and the paper, in my opinion, is now suitable to be published on iJERPH